# ONLINE LEARNING UNDER ADVERSARIAL CORRUPTIONS

## ABSTRACT

We study the design of efficient online learning algorithms tolerant to adversarially corrupted rewards. In particular, we study settings where an online algorithm makes a prediction at each time step, and receives a stochastic reward from the environment that can be arbitrarily corrupted with probability $\epsilon \in [0, \frac{1}{2})$. Here $\epsilon$ is the noise rate the characterizes the strength of the adversary. As is standard in online learning, we study the design of algorithms with small regret over a period of time steps. However, while the algorithm observes corrupted rewards, we require its regret to be small with respect to the true uncorrupted reward distribution. We build upon recent advances in robust estimation for unsupervised learning problems to design robust online algorithms with near optimal regret in three different scenarios: stochastic multi-armed bandits, linear contextual bandits, and Markov Decision Processes (MDPs) with stochastic rewards and transitions. Finally, we provide empirical evidence regarding the robustness of our proposed algorithms on synthetic and real datasets.

## 1 INTRODUCTION

The study of online learning algorithms has a rich and extensive history (Slivkins, 2019). An online learning algorithm makes a sequence of predictions, one per time step and receives reward. The predictions could involve picking an expert from a given set of experts, or picking an action from a set of available actions as in reinforcement learning settings. The goal of the algorithm is to maximize the long term reward resulting from the sequence of predictions made. The performance of such an algorithm is measured in terms of the *regret*, i.e., the difference in the total reward accumulated by the algorithm and the total reward accumulated by the best expert/action/policy in hindsight. Various online learning models have been studied in the literature depending on whether the rewards are generated i.i.d. from some distribution (Gittins, 1979; Thompson, 1933) or are arbitrary (Auer et al., 2001), and whether the reward for all actions is observed at each time step (*full information* setting) (Littlestone & Warmuth, 1994) vs. observing only the reward of the chosen action (*bandit* setting) (Auer et al., 2002; 2001).

In this work, we initiate the study of online learning algorithms with adversarial reward corruptions. Specifically, we focus on the case where the generated rewards are infrequently masked by an adversary and replaced with potentially unbounded corruptions. In doing so, we develop safeguards that minimize the impact of such corruptions on control algorithms operating in an online setting. For example, consider a reinforcement learning agent that interacts with the real world environment to learn a near optimal policy mapping states to actions. For a given state-action pair $(s, a)$ while the true reward distribution may be stochastic, the observed reward associated with $(s, a)$ will have inherent errors due to real world constraints. We would still like to have online learning algorithms that are robust to these errors and have small regret *when compared to the true reward distribution*. These considerations are important in many applications such as routing, dynamic pricing, autonomous driving and algorithmic trading. For example, a classic problem in routing involves choosing the best route in the presence of noise in the ETA estimation for any given route. Similarly, dynamic pricing algorithms need to be robust to adversarial spikes in demand that may lead to unwanted price surges.

To formally study the above scenarios, we consider an online algorithm that proceeds sequentially, where in each step it makes a prediction and receives a reward. With probability $\epsilon$ the observed reward is adversarially corrupted. More specifically, we take inspiration from Huber's contamination

model that has been successfully applied to study various robust estimation problems in unsupervised learning (Huber, 2011; Tukey, 1975; Chen et al., 2018; Lai et al., 2016; Diakonikolas et al., 2019a). Assuming that $P$ is the true distribution of the rewards, in our model the reward at each step is generated from $(1 - \epsilon)P + \epsilon Q$ where $Q$ is an arbitrary distribution. Here $\epsilon < \frac{1}{2}$ is the noise rate. Under this model we design online algorithms with *near optimal regret*, scaling with $\epsilon$, for three important cases: 1) multi-armed stochastic bandits, 2) linear contextual bandits, and 3) learning in finite state MDPs with stochastic rewards.

**Overview of results.** We first consider the setting of stochastic multi-armed bandits. In this scenario there are $k$ arms numbered $1, 2, \ldots k$. In the standard multi-armed bandit model, at each time step, the algorithm can pull arm $i$ and get a real valued reward $r_i$ generated from a normal distribution with mean $\mu_i$ and variance $\sigma^2$.[1] We let $i^*$ represents the best arm, that is, $\mu_{i^*} \geq \mu_i$, $\forall i$. In the $\epsilon$-corrupted model we assume that the reward for pulling arm $i$ at time $t$ is $\tilde{r}_i^t \sim (1 - \epsilon)\mathcal{N}(\mu_i, \sigma^2) + \epsilon Q_t$, where $Q_t$ is an arbitrary distribution chosen by an adversary. We assume that the adversary has complete knowledge of the sequence of predictions and rewards up to time $t - 1$ as well as the true mean rewards and any internal state of the algorithm. Over $T$ time steps, the pseudo-regret of an algorithm $\mathcal{A}$ that pulls arms $(i_1, i_2, \ldots, i_T)$ is defined as

$$\text{Reg}_{\mathcal{A}} = \mu_{i^*} \cdot T - \mathbb{E}[\sum_{t=1}^{T} r_{i_t}]. \tag{1}$$

Notice that while the adversary masks the true rewards with the corrupted ones, we still measure the overall performance with respect to the true reward distribution. While this setting has been studied in recent works (Lykouris et al., 2018; Gupta et al., 2019; Kapoor et al., 2019) they either assume corruptions of bounded magnitude, or provide sub-optimal performance guarantees (see the discussion in Section A). In particular, we provide the following near-optimal regret guarantee based on a robust implementation of the UCB algorithm (Auer et al., 2002).

**Theorem 1** (Informal Theorem). *For the $\epsilon$ adversarially corrupted stochastic multi armed bandit problem, there is an efficient robust online algorithm that achieves a pseudo regret bounded by $\tilde{O}(\sigma\sqrt{kT}) + O(\sigma\epsilon T)$.*

The first term in the above bound is the optimal worst case regret bound achievable for the standard stochastic multi armed bandit setting (Auer et al., 2002). The second term denotes the additional regret incurred due to the corruptions. Furthermore, the work of Kapoor et al. (2019) showed that additional $\sigma\epsilon T$ penalty is unavoidable in the worst case, thereby making the above guarantee optimal up to a constant factor. Furthermore, as in the case of stochastic bandits with no corruptions, we can also obtain instance wise guarantees where the first term above depends on logarithmically in $T$ and on an instance dependent quantity that captures how fare off are the arms as compared to the best one. See Appendix B for details.

Next we consider the case of contextual stochastic bandits. We study adversarially corrupted linear contextual bandits. In the standard setting of linear contextual bandits (Li et al., 2010) there are $k$ arms with $k$ associated (unknown) mean vectors $\mu_1^*, \ldots, \mu_k^* \in \mathbb{R}^d$. At time $t$, the online algorithm sees $k$ context vectors $x_1^t, \ldots x_k^t \in \mathbb{R}^d$, one per arm. If the algorithm pulls arm $i$ then the reward is generated from the distribution $\mathcal{N}(\mu_i^* \cdot x_i^t, \sigma^2)$. In the corrupted setting, we allow at certain time steps, the rewards to be corrupted by an adversary. In particular, we assume that the context vectors are drawn i.i.d. from $\mathcal{N}(0, I)$. Given the true context $x_i^t$, as in the stochastic bandits setting we let the observed reward be generated from $r_i^t \sim (1 - \epsilon)\mathcal{N}(\mu_i^* \cdot x_i^t, \sigma^2) + \epsilon Q_t$ where $Q_t$ is an arbitrary distribution. Given a sequence of arm pulls $(i_1, \ldots, i_T)$ we define the pseudo-regret of an algorithm as

$$\text{Reg}_{\mathcal{A}} = \sum_{t=1}^{T} \mathbb{E}[\max_i \mu_i^* \cdot x_i^t] - \mathbb{E}[\sum_{t=1}^{T} r_{i_t}]. \tag{2}$$

In the above definition, the expectation is again taken over the distribution of contexts, the stochastic rewards and the internal randomness of the algorithm. For this case we provide the following near optimal regret guarantee

---

[1]For simplicity we assume that the variance for each arm distribution is the same. Our results can also be easily extended to handle different variance and also handle the more standard setting where the true rewards are bounded in $[0, 1]$.

**Theorem 2** (Informal Theorem). *For the $\epsilon$ adversarially corrupted linear contextual multi armed bandit problem, there is an efficient robust online algorithm that achieves a pseudo regret bounded by* $\tilde{O}(\sigma d \sqrt{d} k \log(T) \sqrt{T}) + O(\sigma \epsilon \sqrt{d} \log(1/\epsilon) T)$.

As in the case of stochastic bandits, the first term is simply the regret bound for the standard linear contextual bandits problem achieved by the LinUCB algorithm (Chu et al., 2011). The second term is the additional penalty due to the corruptions and is off from the lower bound of $\epsilon \sqrt{d} T$ by a $\log(1/\epsilon)$ factor. Theorem 2 is proved in Appendix C

Finally, we consider the most general setting of learning in Markov Decision Processes (MDPs) under corruptions. We consider a Markov Decision Process (MDP) with state space $\mathcal{S}$, action space $\mathcal{A}$ and transition probabilities specified by $\mathcal{P}$. If an action $a \in \mathcal{A}$ is taken from state $s \in \mathcal{S}$, then the next state distribution is specified as $p(s'|s, a)$. Moreover a stochastic reward $r_{s,a}$ is received where $r_{s,a} \sim N(\mu_{s,a}, \sigma^2)$. The parameters $\mu_{s,a}$ and the transition probabilities are unknown to the learning agent. We assume that the agent start in a fixed state $s_1 \in \mathcal{S}$ and given a policy $\pi : \mathcal{S} \to \mathcal{A}$, follows the trajectory $(s_1, \pi(s_1)), (s_2, \pi(s_2)), \dots, (s_T, \pi(s_T))$. The total reward accumulated over $T$ time steps equals

$$R_\pi = \sum_{t=1}^{T} r_{s_t, \pi(s_t)}. \tag{3}$$

Let $\pi^*$ be the optimal policy defined as $\pi^* = \arg\max_\pi \mathbb{E}[R_\pi]$. Here the expectation is taken over the randomness in the state transitions, the stochastic rewards and the randomness in the policy $\pi$ itself. Given any other policy $\pi$ we define the pseudo-regret of $\pi$ to be

$$\mathsf{Reg}_\pi = \mathbb{E}[R_{\pi^*}] - \mathbb{E}[R_\pi]. \tag{4}$$

In the above setting, the UCRL2 algorithm (Auer et al., 2009) achieves a regret of $\tilde{O}(D|\mathcal{S}|\sqrt{|\mathcal{A}|T})$ where $D$ is the diameter of the MDP. This is almost tight as there is a matching lower bound of $\Omega(\sqrt{D|\mathcal{S}||\mathcal{A}|T})$.

We extend the above basic model with adversarial reward corruptions. In particular, we assume that at each time step $t$, given a state action pair $(s_t, a_t)$, the reward $\tilde{r}_{s_t, a_t}$ observed by the agent is drawn from $(1 - \epsilon)N(\mu_{s_t, a_t}, \sigma^2) + \epsilon Q_t$. Here $\epsilon$ is the noise rate and $Q_t$ is an arbitrary distribution chosen by an adversary. Furthermore, we will assume that the adversary can choose $Q_t$ using complete knowledge of the full history of the learning algorithm up to time $t$. Furthermore, by taking action $a$ from state $s$ at time $t$, the observed transition is generated from the corrupted transition matrix described as $(1 - \epsilon)p(s'|s, a) + \epsilon q_t'(s'|s, a)$, where again $q_t'$ is an arbitrary distribution chosen by an adversary. Our goal in this setting would be to aim for a regret guarantee of $\tilde{O}(D|\mathcal{S}|\sqrt{|\mathcal{A}|T}) + O(\epsilon \cdot T)$. As in the previous two applications, we are interested in designing policies with low regret with respect to the observations from true MDP. For this case we provide the following near optimal guarantee.

**Theorem 3** (Informal Theorem). *For the $\epsilon$-adversarially corrupted MDP model as described above, there is an there is an efficient robust online algorithm $\mathcal{A}$ that achieves a pseudo regret bounded by*

$$\mathsf{Reg}_\mathcal{A} = O(\sigma D|\mathcal{S}|\sqrt{|\mathcal{A}|T \log(|\mathcal{S}||\mathcal{A}|T)}) + O(\sigma \epsilon T).$$

The first term corresponds to the regret achieved by the UCRL2 algorithm (Auer et al., 2009) for the standard MDP setting. The second term is the additional penalty due to corruptions and is again unavoidable in the worst case.

**Techniques.**   Our work combines classical no-regret learning algorithms with recent advances in designing robust algorithms for problems in unsupervised learning. For the case of stochastic multi-armed bandits we modify the standard UCB algorithm (Auer et al., 2002). The UCB algorithm works by maintaining optimistic estimates for the true mean reward of each arm. These optimistic estimates are obtained by using confidence intervals build around the average observed rewards. However, in the presence of adversarial corruptions, these estimates could be arbitrarily bad as we demonstrate in Lemma 1. Instead, we build confidence intervals around the median that is known to be robust to corruptions in Huber's model (Lai et al., 2016; Diakonikolas et al., 2018a). For the case of linear contextual bandits we modify the popular LinUCB algorithm. The LinUCB algorithm

works by maintaining uncertainty estimates around the true mean vectors and picking arms according to these estimates. These estimates are built by solving a least squares problem at each time step. Under adversarial corruptions one would like to build these estimates in a robust manner. However, a straightforward extension of the stochastic bandits case leads to a suboptimal additive penalty of $O(\epsilon dT)$. Instead, we build upon the recent work of Diakonikolas et al. (2019b) for robust high dimensional linear regression to build better uncertainty estimates resulting in the near optimal penalty of $\epsilon \sqrt{d} \log(\frac{1}{\epsilon})T$.

For the case of learning in MDPs, we first consider MDPs with deterministic transition and adversarially corrupted. Here we show that an extension of a UCB style exploration scheme achieves an optimal penalty of $O(\epsilon T)$ by maintaining robust optimistic estimates of rewards at each state-action pair. We then extend this to the more general case where we modify the UCRL2 algorithm (Auer et al., 2009) by maintaining robust estimates of the estimated rewards and transition probabilities.

## 2   STOCHASTIC BANDITS

In this section we consider the setting of stochastic multi armed bandits. Here one has $k$ arms. In each time step, a learning algorithm can pull arm $i$ and observe a reward $r_i$ distributed as $N(\mu_i, \sigma^2)$. Let $i^*$ be the arm with the highest expected reward. Then over $T$ time steps, the pseudo-regret of an algorithm $\mathcal{A}$ that pulls arms $(i_1, i_2, \ldots, i_T)$ is defined as

$$\mathsf{Reg}_{\mathcal{A}} = \mu_{i^*} \cdot T - \mathbb{E}[\sum_{t=1}^{T} r_{i_t}]. \tag{5}$$

There are many algorithms the near-optimal regret of $\tilde{O}(\sigma\sqrt{kT})$ in this setting. The most popular among them is the UCB algorithm (Auer et al., 2002; Slivkins, 2019). When the rewards are in $[0, 1]$, the UCB algorithm works by maintaining optimistic estimates of the average reward seen for each arm. Specifically, the estimate $\gamma_{i,t}$ for arm $i$ at time $t$ is defined as

$$\gamma_{i,t} = \hat{\mu}_{i,t} + 4\sigma\sqrt{\frac{\log kT}{n_{i,t}}}. \tag{6}$$

Here $\hat{\mu}_{i,t}$ is the average reward observed for arm $i$ till time step $t$, and $n_{i,t}$ is the number of times arm $i$ has been pulled up to and including time step $t$. The UCB algorithm starts by pulling each arm once, and then at each time pulling the arm with the best current optimistic estimate as defined in (6).

**UCB can be arbitrarily bad under adversarial corruptions.** We now consider the adversarial model as defined in Section 1 where the $\epsilon$-corrupted rewards $\tilde{r}_i$ are observed each time. In this case it is easy to see that the regret of the UCB algorithm can be arbitrarily bad. This is formalized in the lemma below.

**Lemma 1.** *For any $c > 0$ and $\epsilon \in (\frac{1}{10}, 1)$, there exists a stochastic multi armed bandit setting and an adversary such that the pseudo-regret of the UCB algorithm is at least $c \cdot T$.*

We next show a simple modification of the UCB algorithm that will achieve a near optimal regret of $\tilde{O}(\sigma\sqrt{kT}) + O(\sigma\epsilon \cdot T)$. The algorithm maintains the following optimistic estimates

$$\gamma_{i,t} = \tilde{\mu}_{i,t} + 4\sigma\sqrt{\frac{\log(kT\mu_{\max})}{n_{i,t}}}. \tag{7}$$

Here, $\mu_{\max}$ is an upper bound on the mean rewards, and $\tilde{\mu}_{i,t}$ is defined to be the *median* reward obtained for arm $i$ so far. The robust algorithm is sketched in Figure 1. For the robust UCB algorithm we have the following guarantee.

**Theorem 4.** *Under the adversarially corrupted stochastic bandits model the algorithm in Figure 1 achieves a pseudo-regret of $\tilde{O}(\sigma\sqrt{kT}) + O(\sigma\epsilon T)$.*

*Proof.* Is it known that the median is a more robust estimate than the mean. In particular, the result of Lai et al. (2016) implies that with probability at least $1 - \frac{1}{\mu_{\max}T^4}$, for each arm $i$, and each time

---

**Input:** The $k$ arms, reward variance $\sigma^2$.

1. Play each arm once and update the estimates as in (7).
2. For each subsequent time step $t$, pick the arm $i_t$ with the highest estimate as defined in (7). Play arm $i_t$ and update the estimates.

---

Figure 1: A robust UCB algorithm.

step $t \leq T$, it will hold that

$$|\tilde{\mu}_{i,t} - \mu_i| \leq O(\sigma \cdot \epsilon) + 2\sigma \sqrt{\frac{\log(kT\mu_{\max})}{n_{i,t}}}. \tag{8}$$

Conditioned on the above good event we have that for each time step $t$ and each arm $i$

$$\mu_i - O(\sigma \cdot \epsilon) \leq \gamma_{i,t} \leq \mu_i + O(\sigma \cdot \epsilon) + 6\sigma \sqrt{\frac{\log(kT\mu_{\max})}{n_{i,t}}}. \tag{9}$$

Next, consider an arm $i$ that is pulled $n_{i,t_i}$ times in total, where $t_i$ is the last time step when it is pulled. Then at time $t$ it must hold that

$$\mu_i + O(\sigma \cdot \epsilon) + 6\sigma \sqrt{\frac{\log(kT\mu_{\max})}{n_{i,t_i}}} \geq \gamma_{i,t_i} \geq \gamma_{i^*,t_i} \tag{10}$$

$$= \mu_{i^*} - O(\sigma \cdot \epsilon) \tag{11}$$

$$\Rightarrow O(\sigma \cdot \epsilon) + O\left(\sigma \sqrt{\frac{\log(kT\mu_{\max})}{n_{i,t_i}}}\right) \geq \mu_{i^*} - \mu_i. \tag{12}$$

Hence, conditioned on the good event, the total regret accumulated by playing arm $i$ is

$$n_{i,t_i} (\mu_{i^*} - \mu_i) \leq O(\sigma \cdot \epsilon) n_{i,t_i} + O\left(\sigma \sqrt{\log(kT\mu_{\max}) n_{i,t_i}}\right).$$

Using the fact that $\sum_i n_{i,t_i} \leq T$ with Jensen's inequality and that the good event happens with probability at least $1 - \frac{1}{\mu_{\max} T^4}$, we get that the total pseudo-regret is bounded by $O(\sigma \sqrt{kT \log(kT\mu_{\max})}) + O(\sigma \cdot \epsilon) T$. $\qquad\square$

To contrast the above theorem with the performance of UCB we show the following stonger version of the lower bound in Lemma 1.

**Lemma 2.** *For any $c > 0$ and $\epsilon \in (\frac{1}{10\sqrt{T}}, \frac{1}{\sqrt{T}})$, there exists a stochastic multi armed bandit setting and an adversary such that the pseudo-regret of the UCB algorithm is at least $c \cdot T$ whereas the algorithm in Figure 1 achieves a vanishing regret of $\tilde{O}(\sqrt{kT})$.*

*Proof.* We consider a set of $k$ arms with means $\mu_1, \mu_2, \ldots, \mu_k \in (1, 4c)$ such that the difference between the best arm and all the other arms is at least $2c$. Furthermore, we set the variance $\sigma^2$ in the true Gaussian reward distribution to be $1$. It is easy to see that the robust algorithm will achieve a regret of $\tilde{O}(\sqrt{kT})$ from the guarantee of Theorem 4.

When running UCB, since $\epsilon = \Theta(1/\sqrt{T})$, with high probability the adversary will get to perturb the rewards within the first $O(\sqrt{T})$ time steps. When this happens, the adversary will choose a reward of $-L$ for the best arm and $L$ for all the other arms with $L \gg T$. Hence, no matter which arm the UCB algorithm chooses at that time step, the best arm will always be a suboptimal choice for the algorithm going forward since a large value of $L$ will highly skew the mean reward estimates in the wrong direction for the best arm. As a result, the pseudo regret of UCB will be at least $c \cdot T$. $\qquad\square$

**Input:** The state space $\mathcal{S}$, action space $\mathcal{A}$, reward variance $\sigma^2$.

1. Play each $(s, a) \in \mathcal{S} \times \mathcal{A}$ once and update the estimates as in (20).
2. For episodes $h = 1, 2, \dots$ do:
   - Set start time of episode $h$ to be the current time $t$.
   - For each $(s, a)$ set $v_h(s, a) = 0$.
   - For each $(s, a)$ compute the previous count $N_h(s, a) = \sum_{\tau < t} \mathbb{1}(s_\tau = a, a_\tau = a)$.
   - For each $s, s', a$ compute $\hat{p}_h(s'|s, a)$ using estimates up to time $t$.
   - For each $s, a$ compute $\hat{r}_h(s, a)$ robustly using estimates up to time $t$.
   - Let $M_h$ be the set of all MDPs that whose reward distribution $\tilde{r}$, and transition probabilities $\tilde{p}$ satisfy

$$|\tilde{r}(s, a) - \hat{r}_h(s, a)| \leq 20\sigma \sqrt{\frac{\log(|\mathcal{S}||\mathcal{A}|T\mu_{\max})}{N_h(s, a)}} \tag{13}$$

$$\|\tilde{p}(.|s, a) - \hat{p}_h(.|s, a)\|_1 \leq 20\sigma \sqrt{\frac{|\mathcal{S}|\log(\mathcal{A}|T\mu_{\max})}{N_h(s, a)}}. \tag{14}$$

   - Find the best policy $\pi_h$ that lies in $M_h$.
   - While $v_h(s_t, \pi_h(a_t)) < N_h(s_t, \pi_h(a_t))$
     - choose action $a_t$ according to $\pi_h$ and update the corresponding $v_h$ values. Set $t = t + 1$.

Figure 2: A robust algorithm for general MDPs.

## 3 LEARNING IN MDPS

In this section we study the most general application of our model in learning MDPs under adversarial corruptions. Recall from Section 1 that we consider a Markov Decision Process (MDP) with state space $\mathcal{S}$, action space $\mathcal{A}$ and transition probabilities specified by $\mathcal{P}$. If an action $a \in \mathcal{A}$ is taken from state $s \in \mathcal{S}$, then the next state distribution is specified as $p(s'|s, a)$. Moreover a stochastic reward $r_{s,a}$ is received where $r_{s,a} \sim N(\mu_{s,a}, \sigma^2)$. We will consider a scenario where at each time step, both the reward distribution and the transition matrix is corrupted with an $\epsilon$ probability. We study two settings, one concerning MDPs with deterministic transitions (hence only corrupted rewards) followed by the case of more general MDPs. Due to space constraints we discuss the case of deterministic MDPs in Appendix D.

**Handling General MDPs** To design efficient algorithms for general MDPs. as before we will maintain robust estimates of the rewards and the transition probabilities and use these to guide the search for a near optimal policy. Our proposed algorithm is reminiscent of the UCRL2 algorithm (Auer et al., 2009) and is sketched in the algorithm in Figure 2. For the general case we have the following guarantee. The proof can be found in Appendix E.

**Theorem 5.** *The algorithm from Figure 2 achieves a pseudo regret bounded by*

$$\mathrm{Reg}_{\mathcal{A}} = O(\sigma D |\mathcal{S}| \sqrt{|\mathcal{A}|T \log(|\mathcal{S}||\mathcal{A}|T\mu_{\max})}) + O(\sigma \epsilon T).$$

*Here $\mu_{\max}$ is the maximum mean reward of any state action pair in the MDP.*

## 4 EXPERIMENTS

We empirically validate the robustness of our algorithms to adversarial corruptions. In Section 4.1, we use a real world routing task to benchmark the performance of the robust UCB in Figure 1 when compared to the standard UCB algorithm. Similarly, for the MDP setting, in Section 4.2 we consider routing on randomly generated graphs to compare the performance of the robust UCRL2 in Figure 2 with that of UCRL2. By varying the levels of corruption in the reward structure, we find in both

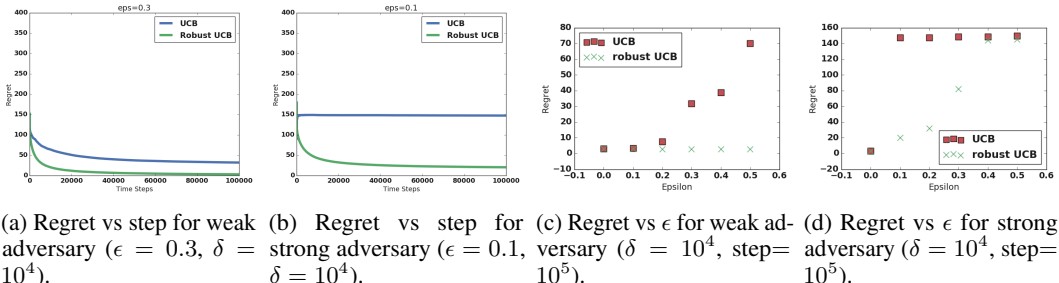

(a) Regret vs step for weak adversary ($\epsilon = 0.3$, $\delta = 10^4$).

(b) Regret vs step for strong adversary ($\epsilon = 0.1$, $\delta = 10^4$).

(c) Regret vs $\epsilon$ for weak adversary ($\delta = 10^4$, step= $10^5$).

(d) Regret vs $\epsilon$ for strong adversary ($\delta = 10^4$, step= $10^5$).

Figure 3: Comparison of vanilla and robust versions of the UCB algorithm.

these settings that the learned policies and the regret incurred are far more resilient under our robust algorithms.

In our experiments, we consider two modes of adversarial corruptions:

- *weak* adversary that corrupts rewards with $U[0, \delta]$ for all actions, and

- *strong* adversary that corrupts rewards with $U[-\delta, 0]$ for the optimal actions and with $U[0, \delta]$ for others.

Here $U[0, \delta]$ denotes the uniform distribution in $[0, \delta]$. Note that a weak adversary shrinks the mean rewards for all actions towards $\delta/2$; this makes the learning harder but otherwise preserves the ranking of actions. A strong adversary on the other hand enhances bad actions and minimizes good ones, hence obfuscating the ranking of actions. For each of these modes, we vary the adversary's strength via probability and magnitude of the corruptions $\epsilon$ and $\delta$, respectively.

## 4.1 ROAD TRAFFIC ROUTING

We illustrate the robustness of algorithms introduced in Section 2. We consider a routing application, where an agent needs to select one of many alternative routes between two locations. We use the road network and link travel times from the New York City Taxi dataset (Donovan & Work, 2017), which contains hourly average travel times on road segments across New York City. We focus on Manhattan for which dense data is available. We first sample $N$ origin-destination pairs and then $K$ alternative routes for each pair. Note that here routes correspond to arms. The competing routes here are computed using a standard algorithm that utilizes a bidirectional Dijkstra search and filters paths for diversity and near-optimality. The distribution of costs for each action is given by observing the costs of the corresponding path in the historical data on every weekday at 9am.

In our experiments, for $N = 200$ origin-destination pairs we considered $K \in [4, 6]$ alternative routes. For each source-destination pair, we form a stochastic bandit problem with the corresponding routes as available arms. We report the average performance across all bandit problems involving multiple source-destination pairs. We study for each adversary mode the effect of corruption probability $\epsilon = 0, 0.1, \ldots, 0.4$ and magnitude $\delta \in [10, 10000]$ on the regret of UCB and its robust counterpart. Figure 3a shows a representative curve of the per step regret as a function of the step count for a weak adversary. Observe the significant improvement offered by robust UCB at each time step. Under a strong adversary, the performance deteriorates for both algorithms, but the robust variant is more resilient. For example, Figure 3b shows one such setting where UCB fails to learn while the robust version learns and the regret asymptotes. Finally, as expected, this pattern continues to hold for both adversary modes for a range of $\epsilon$ and $\delta$; see Figures 3c and 3d.

## 4.2 LEARNING SHORTEST PATHS ON GRAPHS

We next illustrate the robustness of algorithms introduced in Section 3 here. We consider the problem of learning shortest paths on a road network. We cast this problem as an MDP whose reward and transition structure must be learned while minimizing regret.

The road network is modeled as a graph $G = (V, E)$ whose nodes $V$ represent locations and the edges $E$ the links connecting them. The edge costs correspond to the link commute times. Given a destination $t \in V$ while standing at a location $s \in V$, an agent wishes to use that link $e = (s, s') \in E$ which minimizes its overall commute time from $s$ to $t$. That is, $e$ must lie on the shortest path from $s$ to $t$. We cast this as an MDP with the state $(s, t) \in \mathcal{S} = V \times V$ and the action space $\mathcal{A} = \{1, \ldots, A\}$, where $A = \max_{v \in V}$ Outdegree$(v)$. In state $(s, t)$, the first Outdegree$(s)$ actions correspond to taking an edge $(s, s')$, which changes the state to $(s', t)$; the remaining actions are *invalid* and preserve the state. Upon reaching the destination in state $(t, t)$, we choose the next destination $t'$ by cycling through the nodes in a deterministic fashion and randomly sampling a start node $s'$, leading to state $(s', t')$. This ensures that the states in $\mathcal{S}$ are connected and any trajectory

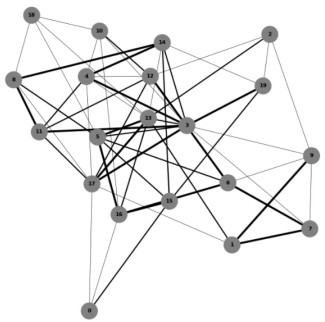

Figure 4: The Erdos-Renyi graph for MDP experiments. The edge thickness indicates its cost.

is infinitely long. Finally, the reward is structured as follows: $N(\mu_G, \sigma^2)$ for optimal actions (e.g. staying on the shortest path), $N(\mu_B, \sigma^2)$ for suboptimal but valid actions, and $N(\mu_H, \sigma^2)$ for invalid actions, where $\mu_G > \mu_B > \mu_H$. Observe that this aligns the agent's objective of maximizing the cumulative reward with finding the shortest path on the original graph $G$.

In our experiments, we use a 20-node Erdos-Renyi graph (Erdős & Rényi, 1960) with edge costs sampled from $\{1, 2, 3\}$ and a maximum outdegree of $10$, as shown in Figure 4. Thus our MDP has $|\mathcal{S}| = 400$ states and $A = 10$ actions in each state, yielding $4000$ state-action pairs that need to be assessed. For the random rewards, we set $\mu_G = 0, \mu_B = -1, \mu_H = -2$ and $\sigma = 1$. Under each adversary mode - weak and strong - the rewards are corrupted with probability $\epsilon = 0, 0.1, \ldots, 0.4$ and magnitudes $\delta \in [10, 10000]$. For each setting, we employ two learning algorithms: the UCRL2 algorithm (Auer et al., 2009) and our robust adaptation in Figure 2. For a rigorous comparison, both implementations share all code except that for computing the empirical reward estimates. For this, in the robust version, the streaming median is estimated using a pool of 10,000 samples updated via reservoir sampling (Vitter, 1985).

Our results indicate that the our algorithm is significantly more resilient to reward corruptions across a wide range of corruption probabilities $\epsilon$ and magnitudes $\delta$. Under a weak adversary, as Figure 5a shows, increasing the frequency of corruption significantly deteriorates UCRL2 performance relative to our robust counterpart. Even so, both algorithms learn near-optimal policies as indicated by the regret values (per step regret near or greater than $1 = -\mu_B$ indicates that learning failed). Increasing the magnitude of corruption, however, completely breaks down the learning for vanilla UCRL2, while the robust version is unaffected; see Figure 5b. Under a strong adversary, the performance of both the algorithms deteriorates but the aforementioned trends continue to hold. UCRL2 fails to learn at $\epsilon = 0.1$, while our robust version has near-optimal performance (Figure 5c). As before though, our robust version continues to learn well under high corruption magnitudes (Figure 5d). In general, we see that that the regret of robust UCRL2 is better at nearly all time steps (Figure 5e). Further, the number of states in which the prescribed action differs from the optimal one is fewer and drops faster (Figure 5f).

## 5 CONCLUSIONS

In this work we initiated the study of robust algorithms for online learning settings. Several open directions come out of our work. It would be interesting to design robust algorithms for linear contextual bandits under more general distribution of context vectors. This would require new algorithms for performing robust regression under more general co-variate distributions.

An important distinction between our proposed robust algorithms and the classical no-regret counterparts is the amount of space usage. We need to store all the rewards (and contexts) observed up to time $t$ to compute good uncertainty estimates. It is an interesting open question to reduce this gap. Finally, it would be interesting to study other scenarios in online learning under adversarial corruptions.

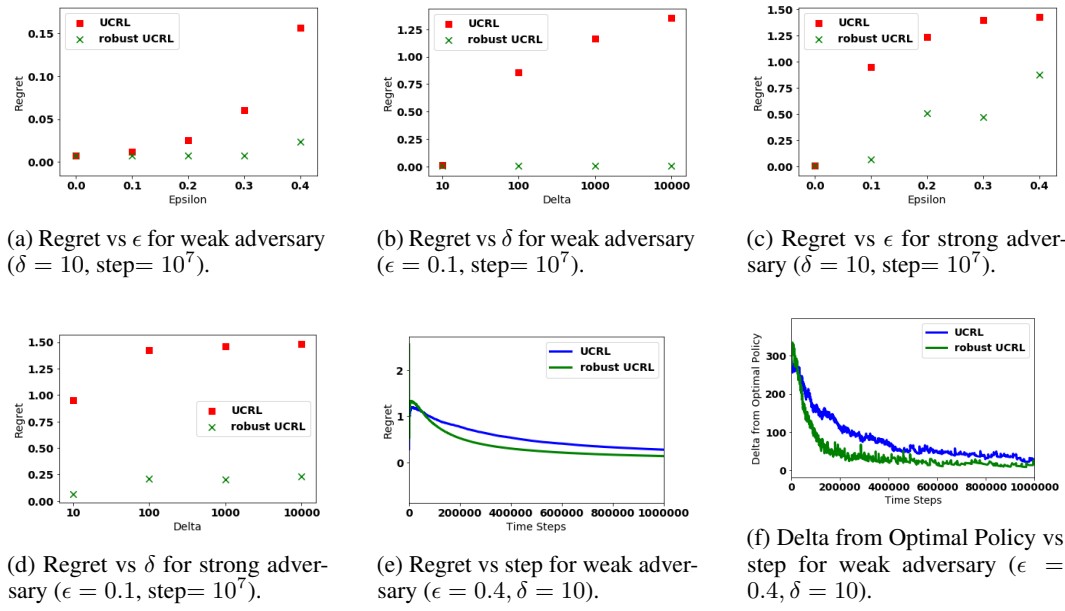

(a) Regret vs $\epsilon$ for weak adversary ($\delta = 10$, step$= 10^7$).

(b) Regret vs $\delta$ for weak adversary ($\epsilon = 0.1$, step$= 10^7$).

(c) Regret vs $\epsilon$ for strong adversary ($\delta = 10$, step$= 10^7$).

(d) Regret vs $\delta$ for strong adversary ($\epsilon = 0.1$, step$= 10^7$).

(e) Regret vs step for weak adversary ($\epsilon = 0.4$, $\delta = 10$).

(f) Delta from Optimal Policy vs step for weak adversary ($\epsilon = 0.4$, $\delta = 10$).

Figure 5: Comparison of vanilla and robust versions of the UCRL2 algorithm.

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
