# OpenReview forum: "Online Learning under Adversarial Corruptions"
_ICLR.cc/2021/Conference — Reject_

### Official Review · AnonReviewer4 · 2020-10-26
**An incremental work which leverages known knowledge and techniques to prove refined regret bounds.**

**Rating:** 5
**Confidence:** 4

**Review:**

The work studies three online learning problems with corrupted rewards as the feedback. The three problems are the stochastic multi-armed bandit, the linear contextual bandit, and the reinforcement learning of the Markov Decision Process optimization.

The major contributions are three improved regret bounds for each of the problems, where the key to success is to replace the empirical mean in the arm/action selection scoring function by the median. Then, robust estimation bounds in the literature are leveraged to achieve the regret analysis. Experiments also support the theoretical findings.

Concerns

Besides the above positive contributions, following are some concerns:

1. The target of optimization. All analyses bound the regrets with respect to the best arm/policy in the uncorrupted situation. This requires more elaboration to connect with the routing example in the introduction. If the actual routing time is the signal to measure the performance, then one does not need to take any ETA estimation into learning. If the actual routing time for the performance evaluation is the time affected by some noise, the regret definition makes no sense. It seems a reasonable situation is that the observations are corrupted but the performance feedback is not. The concern becomes stronger when it comes to the MDPs, as the trajectory of a policy in the corrupted environment will be different from that in the non-corrupted environment. It would be great if the authors can elaborate on the target chosen. Why is regret compared to the uncorrupted situation instead of the corrupted one? How does the trajectory from the corrupted observations affect or not affect the MDP analysis?

2. The key modifications in the algorithms and the key inequalities in the regret analyses are from the literature. The analyses also mainly follow the similar steps in the previous works. Without observing novel technical contributions, the work is considered incremental.

3. There is another work (http://proceedings.mlr.press/v32/seldinb14.html) also dealing with with the mixture of stochastic and the adversarial rewards which is worth mentioning or comparing.

=====================

Post Rebuttal

I went through the authors' reply. My first concern is resolved by the reply. Form the authors' replies to all reviewers, I believe this is an incremental work. It is technically sound, but the lack of involved and novel technical contributions makes it more belong to an incremental work. Thus, I will keep my score unchanged.

---

### Official Review · AnonReviewer3 · 2020-10-28
**Interesting theoretical model and techniques, could use some clarification on MDP algorithm and simulations**

**Rating:** 7
**Confidence:** 4

**Review:**

Summary:
As the title of this paper suggests, the authors look at online learning in three settings of increased generality (stochastic multi-armed bandits, linear contextual bandits and general MDPs), with the added feature that an adversary may corrupt reward (and in the case of MDPs, transition probability) distributions. Corruption may occur on a given turn with probability $\epsilon$, thoguh in general, the degree of corruption an adversary can incur is unbounded. In this model, the authors study regret as the performance metric of their online algorithms, and they provide algorithms for each setting where their regret guarantees can be decomposed into a component which is comporable to state-of-the-art solutions in the no-corruption model, and a second term which depends on the corruption rate, $\epsilon$, and quantified the degree to which ``power'' of the adversary can increase worst-case regret.

To give a brief overview of the algorithmic techniques presented, for the stochastic MAB setting, the authors propose a variant of UCB, whereby the baseline average representing historical performance of an arm is its median rather than the mean of its rewards (which makes sense as a more robust statistic when faced with corruptions), as well as an upper confidence bound similar to UCB. For MDPs, the proposed algorithm maintains a list of feasible policies (those with transitions and rewards ``close'' to what has been observed), and acts optimally according to this feasible set of MDPs (these details are still a bit unclear to me) for a certain number of turns, thus updating the set of feasible MDPs in the process.

Finally, the authors validate their models using simluations on real-world data (road traffic routing in NYC where an agent has to pick a travel route from point A to B, coresponding to arms in the MAB), and on simulated data for the MDP (Erdos Reny Random graphs of 20 nodes where an agent has to find the shortest path of the graph).


Reasons for Score:

I agree with the authors that it is relevant to explore adversarial information production in ML, and this model furthers our knowledge in terms of how to adapt successful online algorithms to adversarial scenarios. Furthermore, the fact that the paper covers online settings of increasing generality is of use in terms of techniques, and indeed the theoretical contributions are interesting in their own right.


Questions / Points of Discussion:
-In general I think it would be great to hear more details on the linear contextual bandits algorithm at the crux of Thm 2. I understand that space is limited, but more details expanding on the line: "Instead we build upon the recent work of Diakonikolas..." could also help transition between the MAB to MDP
-I wonder whether there can be more of a discussion on the dependence of $\mu_{max}$ in your regret bounds.
-I am still a bit confused on the MDP algorithm on page 6. First of all, you mention that $M_h$ is the set of all "close" MDPs to what has been observed, and then $\pi_h$ is called the best policy that "lies in $M_h$". What does this exactly mean? The policy is not an MDP, but I'm guessing it is the best policy against a worst-case MDP from that family of consistent MDPs? This could be clarified a bit
-In relation to this MDP algorithm, I am also curious what its runtime is. It seems to me that it would be computationally expensive to find say the best policy against a worst-case consistent MDP from $M_h$, but this could be clarified as well.
-A final question for the MDP algorithm is also with respect to the initial sampling of all state-action pairs. What does the algorithm do if this is not possible (in a degenerate MDP), or if a single $(s,a)$ is only attainable with extremely low probability?
-As for the simulations, I am curious why such a simple model of adversarial noise was used?:
    -I like that the power of the adversary is bounded by $\delta$, this should be further studied in the theoretical framework
    -On the other hand, the fact that corruptions are either unilateral for all actions, or all but the best action (with a uniform distribution of corruptions), seems to be very limited in the scope of the theoretical framework. Could it be that the algorithms are performing well empirically because of this limited adversarial framework?

---

### Official Review · AnonReviewer1 · 2020-11-02
**New or improved results; writing could be better.**

**Rating:** 5
**Confidence:** 3

**Review:**

This paper studies regret minimizing algorithms for stochastic MABs, linear stochastic bandits, and finite-SA MDPs where epsilon-fraction of rewards/transitions are corrupted by noise of unbounded magnitude. In each case, the resultant regret bound has a epsilon-dependent linear term in T, in addition to standard T^0.5 term. In the latter two settings, the results are the first of their kind; in the first, they improve (in the linear term) over a known result. The general algorithmic approach here is to replace parameter/mean estimation with robust estimators (median, robust regression).

Strengths:
+ The regret bounds are novel, and offer an improvement upon known.
+ The algorithms are fairly natural & modular.

Not ideal:
+ Lemma 1 could be more precise. In the current form, it is unclear if it is better or worse than epsilon * sigma * T. Sigma needs to be large enough for the proof to hold -- so comparisons to c*T remain uncertain. Similarly, at other places, the  writing could be a bit more rigorous.
+ For stochastic linear bandits, typically the regret bound does not scale with the size (or number of vectors) in the decision set. The finitude of the latter is only needed for computational tractability. Why is such a dependence on K necessary here?
+ Finally, could algorithms with tighter regret bounds (as in S^0.5, particularly) be adapted?

To what extent the work here is suitable for ICLR (finite SA MDPs, no func approx); I leave such considerations to the discretion of the other reviewers and the AC. My scores are independent of such aspects.

---

### Official Review · AnonReviewer2 · 2020-11-03
**Review of Paper 2092**

**Rating:** 5
**Confidence:** 4

**Review:**

This paper studies online learning under adversarial corruptions in three settings in the regret minimization setup. The first setting is the classical stochastic multi-armed bandit. The second one is contextual stochastic bandit, and finally, the third is learning in MDPs. In each setting, it is assumed that the rewards (and transition distributions) are corrupted by an adversary with probability $\epsilon$, for some fixed $\epsilon$. For each setting, the paper presents a no-regret algorithm that is robust against such corruption.

Main Comments:

The paper studies interesting online learning problems, which parallels several recent studies on bandits with corruptions. The considered corruption model is however different from the ones in the existing literature.

For each of the studied settings, the paper presents a variant of the classical algorithm for that setting, which is tailored to be robust against $\epsilon$-corruption. For example, for classical stochastic bandits, the paper presents a variant of UCB that achieves a sublinear regret against $\epsilon$-corruption. This is a strong aspect of the paper, as the presented algorithms have similar designs and complexities to their corruption-oblivious counterparts.

The paper is overall well-written and is clearly presented. The paper, however, could still benefit from polishing, and in particular, by fixing typos listed below.

On the technical side, I have some comments:

- For the three settings considered in the paper, it turns out that the price to pay for being robust against $\epsilon$-corruption is an additive penalty of $O(\epsilon T)$ to the regret. This further implies that the interesting regime happens when $\epsilon$ is $O(1/\sqrt{T})$, or decays as $O(1/\sqrt{t})$, since otherwise all the reported regret bounds would scale as $O(T)$ no matter how small a “fixed” $\epsilon$ is. This way, Lemma 1 renders useless as it assumes a fixed $\epsilon$.

- The presented algorithms mildly require some prior knowledge of $\mu_{\max}$. While for bounded rewards, this is no longer the case for, e.g., Gaussian distributed rewards, as $\mu_{\max}$ can be arbitrarily large. This is relevant in particular for the contextual stochastic bandit setting. In addition, for classical stochastic bandits, UCB is presented for Gaussian rewards.


- In the stochastic bandit setting, a median estimator for mean rewards is used instead of the empirical estimates. However, it is not crystal clear to me where in the proof you used the properties of such a median estimator. Is it only in the concentration of good events?

- It would be also tempting to use median estimators in the MDP setting too. Why did you use there only empirical estimates for mean rewards (and transition probabilities?)

- There is a mistake in the analysis for the MDP setup (which is straightforward to fix): In the right-hand side of (31) and (32), a term of $\widetilde {\mathcal O}(\sqrt{T})$ is missing; see the analysis of UCRL2 in (Jaksch et al., 2010).

Overall I do not see the presented results of enough significance to make the paper eligible for ICLR.

Some typos:

p. 2: depend on logarithmically in $T$ => logarithmically depend on $T$

p. 2: let $i^*$ represents => … represent

p. 3: agent start => agent starts

p. 3: from true MDP => … the true MDP

p. 3: confidence intervals build => … built

p. 4: Is it known => It is …

p. 11: A recent of => A recent work of

p. 11: foe => for

p. 12: time spend => time spent

---

### Public Comment · ~Jeffrey_Negrea1 · 2020-11-16
**Additional Related Work**

I enjoyed reading this paper, and found the combination of robust statistical techniques with online learning interesting and compelling.

In addition to the reference suggested by reviewer 4, the literature review could be further improved by discussing some recent work on prediction with corrupted data for both bandits (ZS18) and full-information (Amir+20). Additionally, there is recent literature on the broader subject of robust online learning under various deviations from IID data. (MG18) shows that in the full information setting, one algorithm is adaptively optimal between the adversarial setting and the IID-with-a-gap setting, and almost optimal under relaxations of the IID-with-a-gap setting where there is a single best expert with a gap in expectation. (BNR20) provides an algorithm that adapts minimax optimally under arbitrary, unknown, convex constraint sets on the data generating mechanism in the full information setting. (ZS18) provides a simple stochastic-and-adversarial optimal algorithm for bandits, and extends this to a relaxation of the IID setting where there is a fixed gap between the expected reward of each arm.

MG18: https://arxiv.org/abs/1809.01382
ZS18: https://arxiv.org/abs/1807.07623
Amir+20: https://arxiv.org/abs/2002.10286
BNR20: https://arxiv.org/abs/2007.06552

---

### Decision · Program_Chairs · 2021-01-07
**Final Decision**

**Decision:**

Reject

**Comment:**

The discussion with the expert reviewers reached the consensus that the paper lacks in novel *technical* contributions, and as such it does not meet the bar for a theory-oriented paper at ICLR.